# Divergent national-scale trends of microbial and animal biodiversity revealed across diverse temperate soil ecosystems

Paul B.L. George[1,2], Delphine Lallias[3], Simon Creer [1], Fiona M. Seaton [1,2], John G. Kenny[4], Richard M. Eccles[4], Robert I. Griffiths [2], Inma Lebron[2], Bridget A. Emmett[2], David A. Robinson [2] & Davey L. Jones[1,5]

Soil biota accounts for ~25% of global biodiversity and is vital to nutrient cycling and primary production. There is growing momentum to study total belowground biodiversity across large ecological scales to understand how habitat and soil properties shape belowground communities. Microbial and animal components of belowground communities follow divergent responses to soil properties and land use intensification; however, it is unclear whether this extends across heterogeneous ecosystems. Here, a national-scale metabarcoding analysis of 436 locations across 7 different temperate ecosystems shows that belowground animal and microbial (bacteria, archaea, fungi, and protists) richness follow divergent trends, whereas β-diversity does not. Animal richness is governed by intensive land use and unaffected by soil properties, while microbial richness was driven by environmental properties across land uses. Our findings demonstrate that established divergent patterns of belowground microbial and animal diversity are consistent across heterogeneous land uses and are detectable using a standardised metabarcoding approach.

[1] School of Natural Sciences, Bangor University, Deiniol Road, Bangor LL57 2UW Gwynedd, UK. [2] Centre for Ecology & Hydrology, Environment Centre Wales, Deiniol Road, Bangor LL57 2UW Gwynedd, UK. [3] GABI, INRA, AgroParisTech, Université Paris-Saclay, 78350 Jouy-en-Josas, France. [4] Centre for Genomic Research, University of Liverpool, Crown Street, Liverpool L69 7ZB, UK. [5] School of Agriculture and Environment, University of Western Australia, Crawley, WA 6009, Australia. Correspondence and requests for materials should be addressed to P.B.L.G. (email: afp67e@bangor.ac.uk)

Soil biota, including bacteria, archaea, protists, fungi, and animals, underpin globally important ecosystem functions. Fundamental functions of soil communities include nutrient and hydrological cycling, decomposition, pollution mitigation, and supporting terrestrial primary production, which are inextricably linked to global food security, climate regulation, and other ecosystem services[1,2]. Nevertheless, until recently, characterising soil biodiversity (popularly referred to as a 'black box') has been constrained by our inability to identify typically intractable levels of diversity using either traditional or molecular approaches. High-throughput sequencing has however resulted in a step change, facilitating the characterisation of bacteria[3–7], archaea[6–8], fungi[9,10], protists[11–13], and animals[14] within the belowground biosphere. Increasingly, efforts have been made to investigate the total biodiversity of the soil biosphere across large ecological[15–17] and taxonomic scales[15,16,18,19].

Understanding the response of the total soil biosphere to changes in land use and environmental drivers has become an important research focus in regional soil monitoring programmes[15,16,19] and in small-scale field[20,21] and mesocosm experiments[18,20]. Yet despite the move towards unified study of soil biota, fundamental challenges of technique and scale remain. Often, such studies require the comparison of soil biota metrics captured through both traditional and modern molecular techniques[15,19–21]. To our knowledge, relatively few studies have attempted to assess all components of belowground communities using a multi-marker metabarcoding approach[22].

There is mounting evidence that the microbial and animal fractions of soil communities may respond differentially to land use change. Microbial richness increases[15], whereas richness of soil fauna declines in response to more intense land use[15,23,24]. However, these findings come from relatively homogenous landscapes, such as grasslands[15]. It is unclear whether the differential responses of soil microbes and fauna extend across heterogeneous land uses. For example, across heterogeneous landscapes of Wales, UK, α-diversity of mesofauna is both lowest in agricultural and bog systems, which are the most- and least-intensively managed systems in the country, respectively[23]. Changes in soil properties may further dictate declines of common soil fauna in low-intensity land uses. Therefore, it is critical to assess whether the positive effect of increasing land use intensity on microbial richness is consistent across regions made up of markedly diverse ecosystems and land uses. Similarly, the importance of individual soil properties in shaping belowground communities has also proven difficult to disentangle. Many studies have demonstrated the consistent dominance of pH in shaping belowground community composition at national[23,25–28] and global scales[4,5,9,29]. However, climatic factors[9,30] and other soil properties, including organic matter, nitrogen (N) availability, and the carbon (C)-to-N ratio[9], are also recognised as important drivers of belowground community composition yet consistent trends remain elusive[30]. Therefore it is unclear whether the total soil biosphere responds to changes in land use and soil properties in the same manner across heterogeneous landscapes.

Here, we sought to assess whether divergent responses to land use and soil properties in the microbial and animal fractions of soil communities persist across heterogeneous systems at the national-scale using a standardised metabarcoding approach. We present a national-scale analysis of soil biodiversity across Wales, UK, from the micro-to-macro scale including all major groups of soil microbes in addition to animals, from 436 sites over 2 years across a diverse array of oceanic-temperate ecosystems, including grasslands, forests, bogs, and managed systems. Biotic metrics come from high-throughput sequencing of prokaryotic, fungal, microbial eukaryotic, and soil animal communities using 16S, ITS, and 18S rRNA marker genes; these are complemented by an extensive suite of co-located abiotic soil properties and vegetation cover data. Specifically, we investigate how richness and β-diversity of all major fractions of subterranean life respond to land use type and prevailing soil properties (e.g. organic matter, pH, and N) to explore which lineages play a demonstrable role in determining belowground community structures across large and complex ecological gradients. Our results demonstrate that across a gradient of heterogeneous land uses, richness of soil animals is governed more by land use regime rather than intrinsic soil properties. In contrast, microbial richness is driven by soil properties and demonstrates a largely linear trend of decreasing richness along a productivity gradient of land use based on decreasing soil nutrient availability.

## Results

**Sequencing results.** Illumina sequencing and environmental data were collected from across Wales as part of the Glastir Monitoring and Evaluation Programme (GMEP)[31]. Sample sites were categorised into Aggregate Vegetation Classes (AVCs) based on plant species assessments using established criteria (see Supplementary Note 1). An explanation of the composition of AVCs is described in Supplementary Table 1. Briefly, the 7 AVCs used in the current study were established by clustering samples based on an assessment of vegetation data using a detrended correspondence analysis[32]. The ordination of the detrended correspondence analysis has shown that the land use categories follow a gradient of soil nutrient content[32] from which soil productivity and management intensity can also be inferred (see Supplementary Note 1 and Supplementary Table 1). The AVCs in descending order of productivity are crops/weeds, fertile grassland, infertile grassland, lowland wood, upland wood, moorland grass-mosaic, and heath/bog.

In total, 29,690 bacterial and 156 archaeal operational taxonomic units (OTUs) were identified from 16S reads. Overall, the most abundant class was Alphaproteobacteria (Fig. 1a). Proportional abundances (OTU $n$/total × 100) of Acidobacteria increased in less-productive land use types from its lowest in crops/weeds to its highest in heath/bog AVCs. In contrast, abundances of Actinobacteria followed the exact opposite trend, as did Spartobacteria and Bacilli (Fig. 2a). For archaea, Nitrososphaeria was the most abundant class overall (Fig. 1d); however, the proportion of Thermoplasmata became dominant in less productive AVCs (Fig. 2d).

There were 7582 OTUs recovered from ITS1 sequences. Agaricomycetes were the most abundant class of fungi overall. There was also a large proportion of Sordariomycetes (Fig. 1b). Proportionate abundances of Sordariomycetes and Agaricomycetes followed contrasting trends, with the dominance of the former replaced by the later in lower productivity AVCs (Fig. 2b).

In total, 8683 protist OTUs were recovered from the 18S reads. Chloroplastida (green algae) was by far the most abundant protist group, followed by Rhizaria, Stramenopiles, and then Alveolates (Fig. 1c). Green algae, largely comprised of unidentified sequences (Supplementary Fig. 1a), were least abundant in crops/weed and heath/bog sites (Fig. 2c). Proportions of Rhizaria were relatively constant across AVCs (Fig. 2c) and entirely comprised of Cercozoa (Supplementary Fig. 1b). Among Stramenopiles, proportions of Ochrophyta were also largely consistent, while those of Oomycetes and Bicosoecida followed contrasting trends across the productivity gradient of AVCs, declining and increasing, respectively (Supplementary Fig. 1c). Ciliates were the most common Alveolates in most AVCs; however, the proportion of Apicomplexa was greater in the lowland wood and grassland AVCs (Supplementary Fig. 1d). The proportion of Amoebozoa was surprisingly low (Fig. 1c), potentially due to

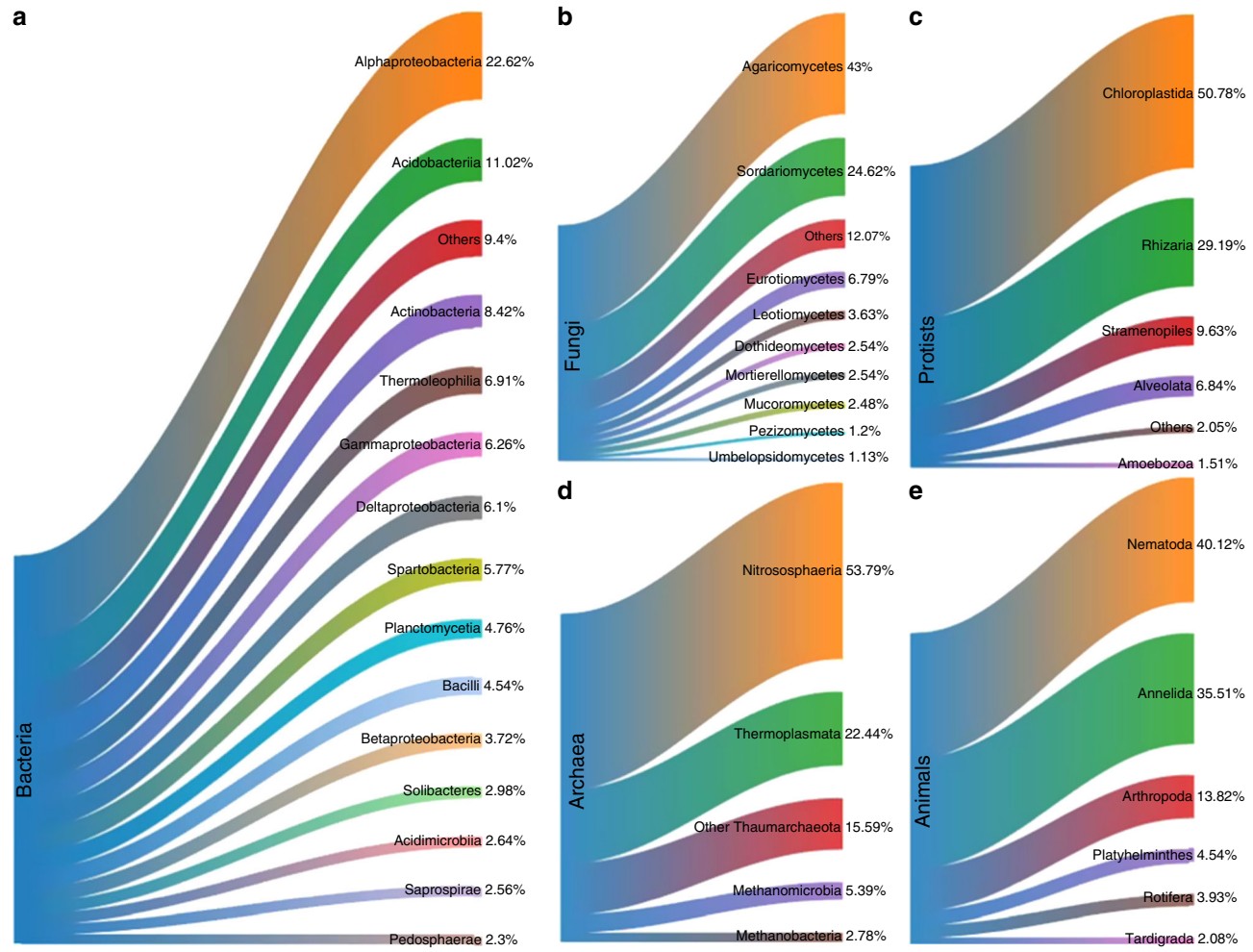

**Fig. 1** Sankey diagrams of proportional abundances of OTUs from all samples for major soil biota groups. Arms denote proportions of OTUs at the class-level for **a** bacteria; **b** fungi; of major lineages of **c** protists; class-level for **d** archaea; and at the phylum-level for **e** animals. For information on how this figure was created, please see Supplementary Methods

primer bias in our study when compared to other studies[12,15]. Across AVCs Tublulinea was consistently dominant among the Amoebozoa, though divergent trends in Gracilipodida and Discosea can be seen along the productivity/intensity gradient (Supplementary Fig. 1e).

In the animal dataset, 1138 OTUs were recovered. Nematode OTUs were the most abundant animal group across all samples (Fig. 1e). Annelids and arthropods followed opposing trends in proportionate abundance, increasing and decreasing respectively, across the productivity gradient. Proportions of Platyhelminthes and Tardigrades also increased in less-productive AVCs (Fig. 2e).

**Effect of land use on belowground richness**. We found significant differences in biodiversity trends across land use types. There was a marked shift along the productivity gradient of crops/weeds-to-heath/bog in all organismal groups, except animals (Fig. 3). Significant differences in the mean richness of bacterial OTUs were prominent ($F_{6,264} = 78.47$, $p < 0.0001$) following ANOVA. Bacterial richness decreased in AVCs across the productivity gradient with highest values in the most productive crops/weeds and grasslands and lowest in the low productivity land uses (i.e. moorland grass-mosaic, heath/bog) (Fig. 3a). The same trend was also observed in fungi ($F_{6,248} = 48.98$, $p < 0.001$;

Fig. 3b), and protists ($F_{6,249} = 59.86$, $p < 0.001$; Fig. 3c). For individual pair-wise comparisons see Supplementary Note 4. Richness of archaeal OTUs had an opposing trend to that of other microbial groups. Archaeal OTU richness was significantly lower ($F_{6,185} = 24.37$, $p < 0.001$) in higher-productivity AVCs and highest in the least-productive land-use types (Fig. 3d). In the crops/weeds, AVC richness of archaeal OTUs was significantly lower than upland wood ($p = 0.01$), moorland grass-mosaic ($p = 0.005$), and heath/bog sites ($p < 0.001$) based on Tukey's post hoc tests, with the remaining land uses displaying intermediate OTU richness values.

Animal OTU richness did not follow the trends observed in microbial communities. Differences observed with ANOVA were significant ($F_{6,244} = 6.25$, $p < 0.001$) but plateaued after the grassland AVCs, as opposed to the sloped trend of microbial groups across the productivity gradient (Fig. 3e). Richness in the infertile grasslands was significantly greater than in crops/weeds ($p = 0.008$), heath/bog ($p = 0.003$), and upland wood ($p = 0.02$) based on Tukey's post hoc tests. Richness was lowest in the most intensively management crops/weeds sites and was shown to be significantly lower than richness of lowland woods ($p = 0.04$) with Tukey's test. Collectively, the results demonstrate a strong divergence between the richness of animal and microbial communities across all AVCs.

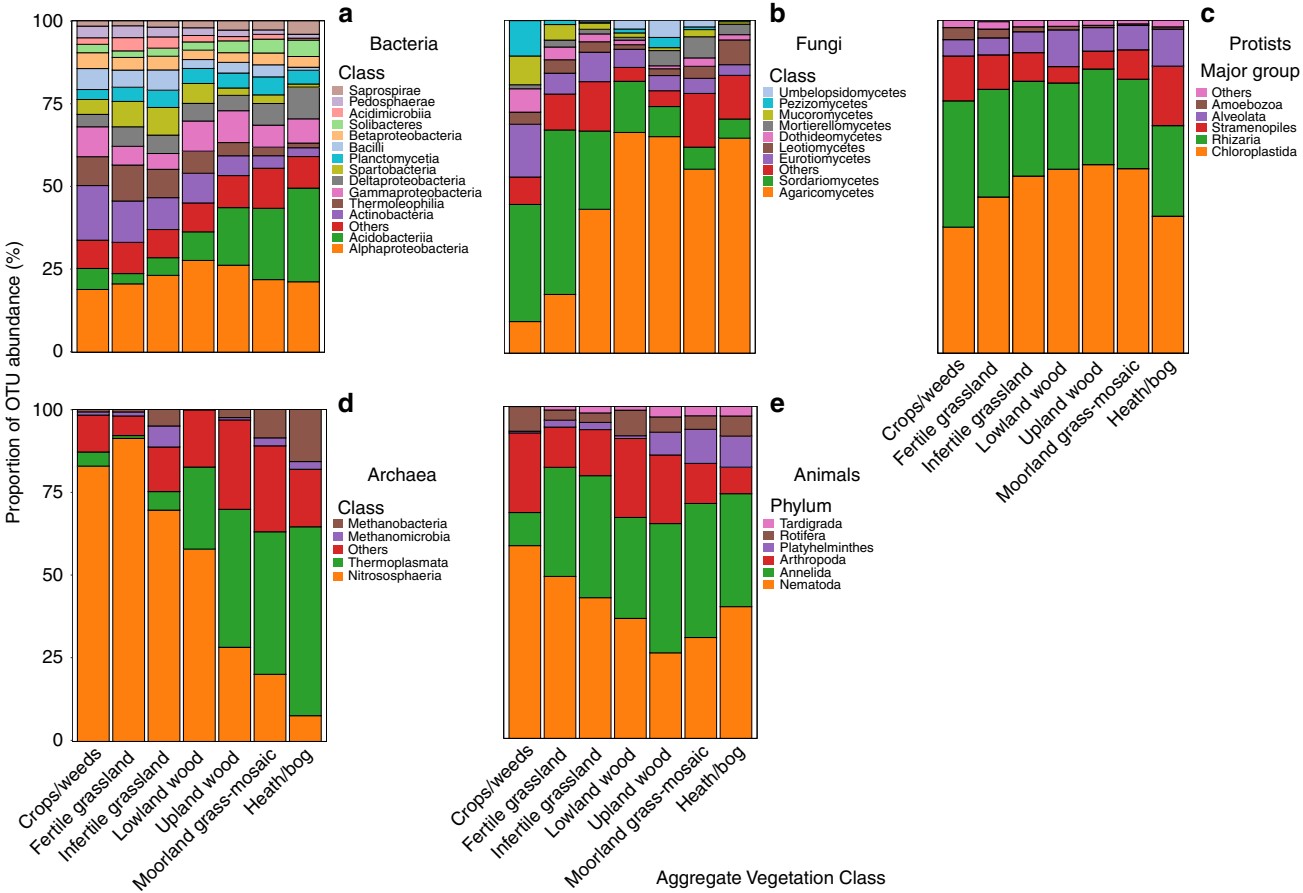

**Fig. 2** Proportionate abundances of OTUs for major soil biota groups within each Aggregate Vegetation Class. Land uses are ordered from most (crops/weeds) to least (heath/bog) using the same divisions as Fig. 1 for **a** bacteria; **b** fungi; **c** protists; **d** archaea; and **e** animals

**Relationships of richness between organismal groups**. Bacterial richness from the total dataset was significantly correlated with all other organismal groups (Supplementary Table 2). Such relationships were positive between bacterial richness and richness of fungi, protists, and animals. Similarly, there was a positive relationship between protistan richness and both fungal and animal richness. However, archaeal richness demonstrated significant, but negative correlations with all organisms except animals. Indeed animal richness (measured by metabarcoding) was only significantly correlated with animals (measured by taxonomic assessment; Table 1) and protists (Supplementary Table 2).

**Relationships between richness and environmental variables**. Partial least squares (PLS) regressions demonstrated that the divergence observed between animal and microbial communities may be due to the effects of soil properties. No soil properties were significantly correlated with richness of soil animal OTUs (Table 1). Conversely, there were strong relationships between microbial richness and a range of soil properties. However, although microbes were influenced by the same environmental variables, there were distinct patterns within each group. For example, while pH was the best predictor of bacterial richness, it was ranked as second for fungi and protists and third for archaea. Bulk density and C:N ratio were also major drivers of richness across all microbial groups. Elevation (here closely linked with precipitation and organic matter content) was the most important environmental variable in relation to archaea and protist richness. Organic matter and bulk density were strong predictors of fungal

OTU richness. All environmental properties that had positive relationships with OTU richness of bacteria, fungi, and protists had negative relationships with archaea.

**Community structure (β-diversity) across land uses**. Non-metric multidimensional scaling (NMDS) using Bray–Curtis distances showed consistent differences in β-diversity between AVCs across all organismal groups. Plots show tight clustering of the crops/weeds, fertile grassland, and infertile grassland AVCs, whereas the other AVCs form a more dispersed organismal assemblage (Fig. 4 for bacteria and Supplementary Figs. 2–5). Results of PERMANOVAs were significant across all groups and analyses of dispersion were also significant (Fig. 4 for bacteria and Supplementary Figs. 2–5) for all groups except for the dispersion of animals ($F_{6,401} = 0.67$, $p = 0.68$) owing to the wide range of sample numbers within each AVC (Supplementary Fig. 5). We also found that this clustering was present using constrained canonical analyses of principle components (CAP) ordinations for each organismal group (Supplementary Figs. 6–10).

pH was the best predictor of β-diversity from linear fitting for all soil organisms (Table 2 and Supplementary Tables 3–6). The carbon-to-nitrogen (C:N) ratio was the second most important variable in all major groups except animals. Mean C:N values were higher in the crops/weeds and grassland AVCs and lower in the remaining land use types (Supplementary Table 6). Mean pH values and C:N ratios (Supplementary Table 6) reflect the distribution of points in NMDS plots, with tight groupings observed in the crops/weeds and grasslands AVCs and

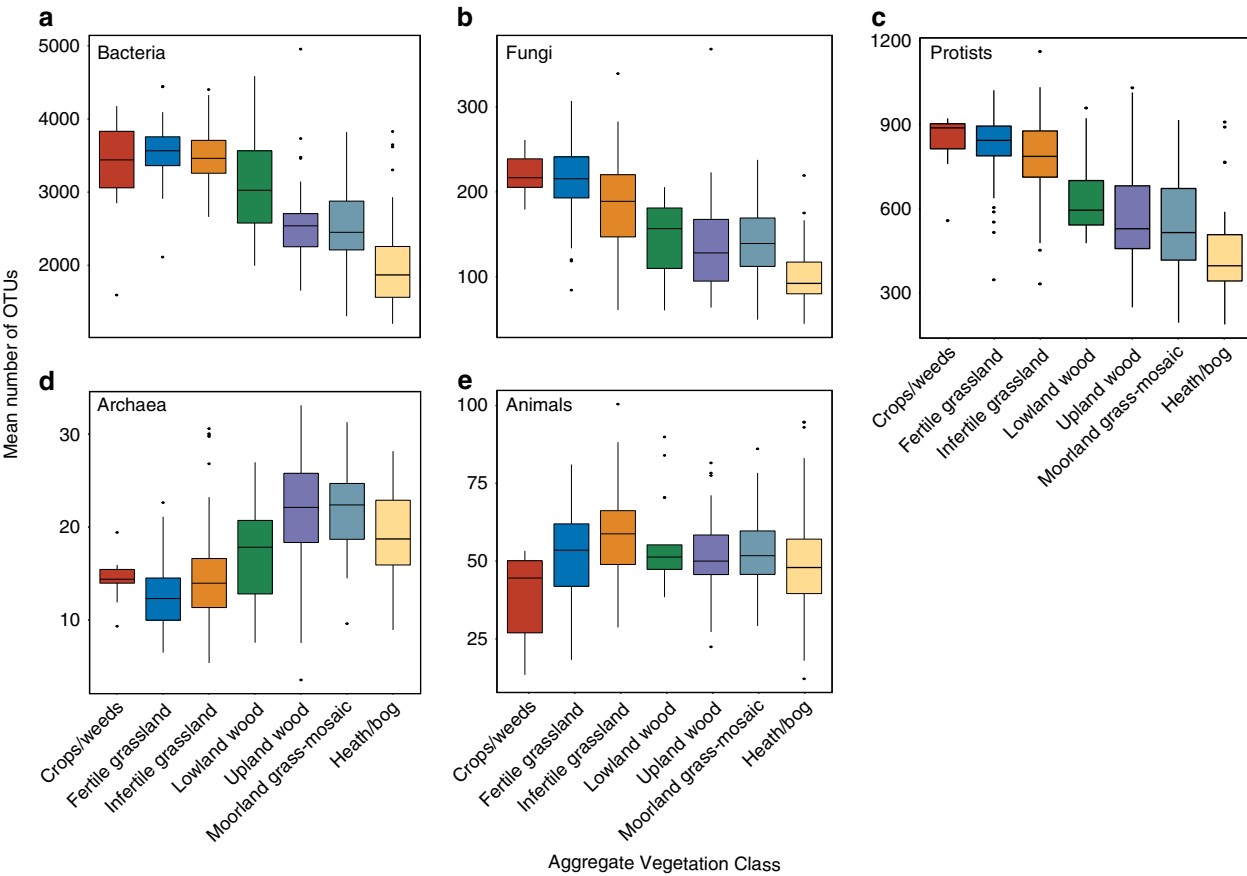

**Fig. 3** Boxplots of OTU richness for each organismal group. Richness of **a** bacteria; **b** fungi; **c** protists; **d** archaea; **e** animals are plotted against Aggregate Vegetation Class ordered from most (crops/weeds) to least (heath/bog) productive. Boxes are bounded on the first and third quartiles; horizontal lines denote medians. Black dots are outliers beyond the whiskers, which denote 1.5× the interquartile range. Source data are provided as a Source Data file

increasingly more spread out groupings in all other AVCs as pH values decreased and became more varied (Fig. 4 for bacteria and Supplementary Figs. 2–5). Across all groups, all or nearly all variables were significant following linear fitting; however, most were only weakly correlated with β-diversity values. Other important variables varied in their ranked importance, including elevation, mean annual precipitation, organic matter content, total C, bulk density, volumetric water content, and clay content of soil (Table 2 and Supplementary Tables 3–6). The results of linear model fitting for CAP ordinations, though not identical (Supplementary Tables 7–11), were highly related to those of the NMDS ordinations (Supplementary Fig. 11).

## Discussion

High-throughput sequencing of the biosphere amongst heterogeneous soils revealed both expected and novel relationships between soil organisms and environmental drivers. The richness of microbes and animals had notable contrasting trends across land use types. The richness of microbial communities was strongly influenced by both land use and environmental variables, especially pH, C:N ratio, elevation, organic matter, and annual precipitation. Conversely, we found no significant associations between measured environmental variables and animal richness, which was negatively impacted by higher intensity land use, suggesting that richness patterns of microbial and macrobial life fractions adhere to different ecological determinants. For β-diversity, pH was by far the most important environmental variable in shaping community

composition of all organismal groups, yet other drivers were attributable for influencing patterns of α-diversity.

Our findings demonstrate that diverging trends between soil microbes and fauna extend across distinct, heterogeneous land uses. Furthermore, we build on the work of Gossner et al.[15] by demonstrating that microbial richness, with the exception of archaea, increases with greater land use intensity across heterogeneous ecosystems at the national-scale. The divergence between microbes and animals at this scale is supported by previous findings from French soils[17,25]. Across France, bacterial richness[17] and biomass[25] were strongly linked to belowground environmental properties but largely unaffected by aboveground climatic variables, which commonly influence animal and plant biogeography[25,30]. Our findings show that richness of fungi and protists also follow this trend—whereas archaea follow an opposing trend to all other groups.

There are several mechanisms that may explain the relationship between higher microbial richness and intensifying anthropogenic disturbance. One explanation is that consistent nutrient inputs from fertilisers and disturbance under tillage stimulate high α-diversity in these areas[16]. Indeed higher α-diversity has been observed in cropping systems than in forest or grassland sites for both bacteria[16,17] and fungi[16]. Interestingly, high microbial richness in more productive land use types (e.g. arable) may illustrate the intermediate disturbance hypothesis (IDH) within soil ecosystems. Under the IDH, as outlined by Connell[33], diversity reaches its highest levels where succession has been interrupted by intermittent disturbance events. In our sites,

**Table 1 Results of partial least squares regressions for soil biota against soil properties for richness**

| Soil and environmental variables | Taxon | | | | |
|---|---|---|---|---|---|
| | **Bacteria** | **Archaea** | **Fungi** | **Protists** | **Animals** |
| Total C[a] | *1.14* ($R^2 = 0.44$***) | **1.21** ($R^2 = 0.13$***) | 0.44 | *1.3* ($R^2 = 0.35$***) | 0.9 |
| Total N[a] | 0.93 | 0.89 | 0.93 | 0.8 | 1.18 |
| C:N ratio[b] | *1.45* ($R^2 = 0.41$***) | **1.31** ($R^2 = 0.09$***) | *1.64* ($R^2 = 0.28$***) | *1.67* ($R^2 = 0.35$***) | 0.1 |
| Total P (mg kg⁻¹)[b] | 0.35 | 0.59 | 0.7 | 0.85 | 0.67 |
| Organic matter (% LOI)[a] | *1.47* ($R^2 = 0.5$***) | **1.27** ($R^2 = 0.14$***) | *1.13* ($R^2 = 0.29$***) | *1.27* ($R^2 = 0.35$***) | 1.08 |
| pH (CaCl₂) | **1.98** ($R^2 = 0.51$***) | *1.68* ($R^2 = 0.25$***) | **1.52** ($R^2 = 0.23$***) | **1.56** ($R^2 = 0.33$***) | 0.9 |
| Soil water repellency[a,c] | *1.31* ($R^2 = 0.2$***) | 0.9 | *1.23* ($R^2 = 0.13$***) | 0.93 | 0.98 |
| Volumetric water content (m³ m⁻³) | 0.36 | **1.33** ($R^2 = 0.13$***) | 0.6 | 0.41 | 0.4 |
| Soil bound water (g water g dry soil⁻¹) | *1.25* ($R^2 = 0.41$***) | 0.83 | *1.08* ($R^2 = 0.26$***) | *1.23* ($R^2 = 0.31$***) | 0.63 |
| Rock volume (mL) | 0.25 | 0.61 | 0.64 | 0.27 | 1.3 |
| Bulk density (g cm⁻³) | **1.39** ($R^2 = 0.44$***) | *1.43* ($R^2 = 0.18$***) | **1.41** ($R^2 = 0.29$***) | **1.5** ($R^2 = 0.35$***) | 1.39 |
| Clay content (%)[d] | 0.85 | *1.19* ($R^2 = 0.1$***) | 0.84 | **1.14** ($R^2 = 0.09$***) | 0.05 |
| Sand content (%)[d] | 0.45 | 0.16 | 0.6 | 0.51 | 0.78 |
| Elevation (m) | *1.66* ($R^2 = 0.42$***) | **1.7** ($R^2 = 0.27$***) | *1.68* ($R^2 = 0.22$***) | *1.65* ($R^2 = 0.36$***) | 0.57 |
| Mean annual precipitation (mL) | *1.08* ($R^2 = 0.25$***) | **1.75** ($R^2 = 0.3$***) | *1.44* ($R^2 = 0.18$***) | *1.48* ($R^2 = 0.27$***) | 0.46 |
| Temperature (°C) | 0.51 | 0.5 | 0.56 | 0.58 | 0.35 |
| Collembola[e] | 0.34 | 0.06 | 0.41 | 0.17 | **1.14** ($R^2 = 0.03$***) |
| Mites[e] | 0.49 | 0.2 | *1.17* ($R^2 = 0.03$***) | 0.23 | **1.74** ($R^2 = 0.08$***) |
| Total mesofauna[e] | 0.44 | 0.1 | *1.03* ($R^2 = 0.01$*) | 0.15 | **1.71** ($R^2 = 0.08$***) |

Positive relationships are written in bold and negative relationships are written in italics
[a]Log₁₀-transformation
[b]Square-root-transformation
[c]Soil water repellency was derived from median water drop penetration times (s)
[d]Aitchison's log-ratio transformation
[e]Log₁₀ plus 1 transformation
***$p < 0.001$; **$0.001 > p < 0.01$; *$0.01 > p < 0.05$, and blank indicates $p > 0.05$

microbial richness was highest in AVCs concurrent to disturbances (augmented by nutrient inputs) from agricultural interventions such as fertilisation, tilling, clearing, and the cultivation of livestock. However, it is also possible that the high diversity observed in the grassland and especially in agricultural land uses stems from organisms that have entered a dormant state after disturbance-induced changes to their environment[13,34]. Disturbance pressures can also lead to high bacterial diversity through the reduction in dominant OTUs, which are replaced by a wide range of weaker competitors. It has been demonstrated that α-bacterial diversity is greater in the phyllosphere of ivy in urban habitats associated with more anthropogenic stressors than in less disturbed sites[35]. Our findings suggest that the phenomenon of greater species richness resulting from the addition of nutrients and non-equilibrium dynamics induced by disturbance may extend to across all microbial groups, with the possible exception of archaea.

Richness of all microbial groups, except archaea, followed the land use productivity/management intensity gradient[32] with higher richness in the highly productive and more disturbed grasslands and arable sites and lower richness in the least productive, relatively undisturbed upland heath/bog sites. Changes within bacterial and fungal communities reflected expected within-community changes following the shift in soil nutrient quality across land uses. Actinobacteria[36] and Sordariomycetes[37] are known to dominate bacterial and fungal communities in high productivity grasslands as witnessed here. In contrast, Acidobacteria increased in proportion in low productivity, highly acidic AVCs as expected based on previous studies from the UK[27] and across the globe[7]. Likewise, the greater proportion of Agaricomycetes OTUs in low productivity AVCs is intuitive as many Agaricomycete fungi are common in bogs and related low-productivity habitats across Wales[38].

Protists have been chronically overlooked in European soil monitoring programmes (but see ref. [28]), as extracting trends of protist diversity across land uses is difficult. For example, Gossner et al.[15] were not able to show changes in richness across all protists with land use intensification. We demonstrate that protistan richness follows the trends of bacteria and fungi across land uses, with the highest richness levels in arable land. As with other microbes, there is evidence of increased protist richness at the mesocosm[39] and field[40] level, in response to fertiliser addition. Furthermore, in German grassland soils, protist richness has been shown to increase with land use intensity[41]. Our results show that

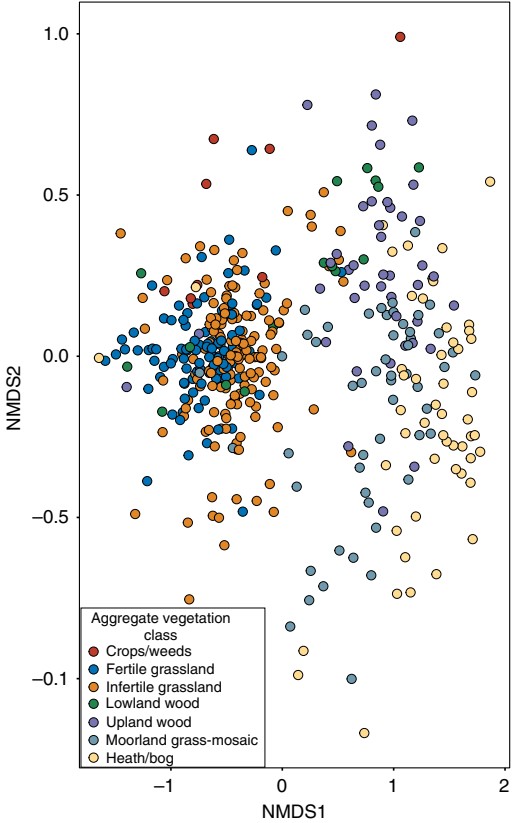

**Fig. 4** Plot of the non-metric dimensional scaling ordination (stress = 0.06) of bacterial community composition across GMEP sites. Samples are coloured by Aggregate Vegetation Class. Results of PERMANOVA ($F_{6,427} = 30.76$, $p = 0.001$) and dispersion of variances of groups ($F_{6,427} = 10.97$, $p = 0.001$) were significant

**Table 2 Summary of relationships amongst environmental factors and bacteria communities**

| Soil and environmental variables | $R^2$ | Correlation | | |
|---|---|---|---|---|
| | | Axis 1 | Axis 2 | Axis 3 |
| pH (CaCl$_2$) | 0.71*** | − | − | + |
| C:N ratio[a] | 0.52*** | + | − | + |
| Volumetric water content (m$^3$ m$^{-1}$) | 0.49*** | + | − | + |
| Bulk density (g cm$^3$ $^{-1}$) | 0.47*** | − | + | − |
| Organic matter (% LOI)[b] | 0.46*** | + | − | + |
| Elevation (m) | 0.45*** | + | − | − |
| Mean annual precipitation (mL) | 0.43*** | + | − | − |
| Total C[b] | 0.39*** | + | − | + |
| Clay content (%)[c] | 0.33*** | − | + | − |
| Soil bound water (g water g dry soil$^{-1}$) | 0.31*** | + | − | + |
| Soil water repellency[b,d] | 0.27*** | + | − | − |
| Total N (%)[b] | 0.26*** | + | − | + |
| Sand content (%)[c] | 0.21*** | + | + | + |
| Collembola[e] | 0.09*** | − | + | − |
| Mites[e] | 0.06*** | + | + | − |
| Total P (mg kg$^{-1}$)[a] | 0.06*** | − | − | − |
| Total mesofauna[e] | 0.06*** | + | + | − |
| Rock volume (mL) | 0.05** | − | + | + |
| Temperature (°C) | 0.03* | + | + | − |

+/− signify the direction of association between each variable and respective NMDS axes
[a]Square-root-transformation
[b]Log$_{10}$-transformation
[c]Aitchison's log-ratio transformation
[d]Soil water repellency was derived from median water drop penetration times (s)
[e]Log$_{10}$ plus 1 transformation
***$p < 0.001$; **$0.001 > p < 0.01$; *$0.01 > p < 0.05$, and blank indicates $p > 0.05$

an association between intensification and protistan richness extends across the national-scale over multiple land uses.

Unlike other microbes, archaeal richness was greatest in low productivity AVCs and lowest in highly productive sites (Fig. 3d). Furthermore, our understanding of the extent of soil archaeal diversity and its functional capabilities is continually increasing[6–8]. Recent research has revealed many lineages of Thaumarchaeota are crucial links in the N cycle and methanogenesis in soils[7,8]. Archaeal richness was highest in the moorland grass-mosaic and heath/bog AVCs, likely due to the specialised nature of acidophilic lineages. In particular, the Thaumarchaeota[42] and Thermoplasmata[43] are known to proliferate (Fig. 2d) under reduced competition from bacteria.

Animal richness did not change linearly with land use and was not strongly influenced by environmental variables. Our molecular analysis of soil eDNA supports recent findings by George et al.[23] based on morphological assessments of coincident soil mesofauna. Both the present work and George et al.[23] demonstrated that animal richness and abundance were lowest in land uses associated with more intensive management. Animal richness peaked in infertile grasslands and was lowest in crops/weeds sites (Fig. 3e). Agricultural disturbance negatively affects soil faunal richness and diversity across large geographic scales[14,23,24]. However, in the low-productivity land uses, although proportional abundances of arthropod taxa declined similarly to the findings of George et al.[23], overall richness was not as strongly affected due to an increase in fractions of Annelids, Platyhelminthes, and Tardigrades. Such an increase in the peat-rich,

low-disturbance, higher elevation sites is rather intuitive since Annelids, Platyhelminthes, and Tardigrades are susceptible to desiccation and require moist habitats to be active components of the soil community[44,45]. As soil animals still exhibited expected lower diversity trends in more intensively managed land uses[15,23,24], there are further opportunities for research into understanding the mechanisms underlying the divergent richness trends between microscopic animals and the rest of soil communities.

Soil pH, as evidenced by ordination results, was the most important environmental variable in our study for β-diversity and in most cases richness as has been previously observed across the UK[27,28] and at larger national[25,26] and continental scales[4–6]. pH has been implicated with driving richness of soil Archaea[42,43] and is the most important driver of protist communities in the UK[28]. However, pH only plays a marginal role in shaping soil protist communities globally[11]. Likewise, pH is a poor predictor of global fungal biogeography, yet is a good predictor of ectomycorrhizal fungal richness[9], which may contribute to the Agaricomycetes OTUs observed in the present study. Nevertheless, it is important to acknowledge the inconsistent nature of correlations between microbial biodiversity and pH, potentially due to variations in soil properties occurring at scales that do not align with large-scale soil surveys[30].

We also observed a strong effect of C:N ratio in determining richness of microbes and β-diversity of all organismal groups, as has been observed in bacterial[27] and protistan[28] β-diversity across Britain and some fungi globally[9]. Yet C:N ratio is often co-correlated with other soil properties including bulk density, total C, organic matter, elevation, and mean annual precipitation. Disentangling such related variables is difficult; despite using PLS analyses[46] we could not disentangle co-correlated soil properties.

For example, AVCs such as moorland grass-mosaic and heath/bog generally had higher elevation, mean annual precipitation, C:N ratio, and both total C and N (Supplementary Table 12) owing to their less-disturbed, upland location, and often peat-rich soils. Higher C:N ratios are indicative of lower-quality soils[47] and have historically been associated with a shift in microbial biomass from bacterial to fungal dominance[48]. Our results suggest that, with the exception of archaea, microbial richness is equally susceptible to the effect of soil quality degradation. According to our results, archaea, on the contrary, appear to be well adapted to habitats with lower nutrient quality.

We observed strong relationships between soil properties and microbial, but not animal richness. We suspect this is due to the direct effects of soil properties on microbes. For example, shifts in pH towards either a more alkaline or acidic condition inhibit the ability of most non-specialised bacteria to uptake nutrients from their environment[26]. In addition the quality of soil nutrients, as discussed previously, was likely a strong determinant of available nutrient resources and therefore total richness of microbes. We also found strong relationships between soil properties and β-diversity and across all organismal groups. These relationships between Bray–Curtis dissimilarities and soil properties demonstrate that more dissimilar belowground communities correlate positively with indicators of better quality soils across the breadth of soil biota (Supplementary Table 6). However, associations between nutrient quality and animal community composition are likely the result of nutrients influencing the composition of the aboveground plant community[49] rather than direct interactions with animals. Furthermore, animals are more vagile than microbes and can actively seek out microhabitats with better resources[50], limiting the direct impact of soil properties on animal richness.

Using an extensive soil sampling programme and metabarcoding, we present perhaps the most comprehensive assessment of the belowground diversity in Europe. Despite uncertainties on the ability of environmental DNA methods using small soil volumes to accurately characterise communities of larger organisms[51], we were still able to detect key differences in larger organisms (i.e. animals) across land uses. Our results highlight the complexity of belowground ecology by demonstrating a divergence of patterns of richness between soil fauna and microorganisms at a national-level. We show that microbial richness is strongly influenced by soil properties in a near-uniform manner, whereas animal richness is not. Rather, animal richness is likely driven by changes in aboveground communities that stem from intensive land use management, while microbial richness was affected by soil properties in addition to land use. A particularly interesting outcome of our analyses is the near-uniform trend of declining microbial richness along a gradient of decreasing land use productivity/management intensity. The data therefore suggest that soil properties strongly affect bacteria, fungi, and protists in a similar manner, whereby richness decreases with soil quality; whereas archaea showed an opposing trend with increasing richness as productivity declined. The richness of animal OTUs, on the contrary, was not affected by soil properties although β-diversity was. Although often considered as ecological 'black boxes', soils continue to provide unique and coherent insights into the differences between interconnected microbial and macrobial assemblages. Our findings also highlight the importance of the dynamics between biotic and abiotic processes that drive the organisation of belowground biological diversity.

## Methods

**Sampling**. Soil samples were collected between late spring and early autumn in 2013 and 2014 as part of GMEP (Supplementary Note 2), established to monitor the Welsh Government's agri-environment scheme, Glastir. The scheme covered an area of 3263 km$^2$ with 4911 landowners[31]. Briefly, surveyors collected samples from randomly selected 1 km$^2$ squares with up to 3 locations within squares, following protocols established by the UK Countryside Survey[52]. As described previously, habitat within plots was classified using plant species assessments into one of seven AVCs[32]: crops/weeds ($n = 9$), fertile grassland ($n = 98$), infertile grassland ($n = 162$), lowland wood ($n = 17$), upland wood ($n = 44$), moorland-grass mosaic ($n = 54$), and heath/bog ($n = 52$) (Supplementary Note 1; Supplementary Table 1). Soil type was derived from the National Soil Map[53] (Supplementary Note 3; Supplementary Table 13). Organic matter content was classified by loss-on-ignition (LOI) following the protocols of the 2007 Countryside Survey[51].

A total of 436 cores were collected from 1 km$^2$ squares, with up to 3 samples coming from an individual square based on a randomised sampling design. Cores were transported to the Centre for Ecology and Hydrology, Bangor, UK, and stored at −80 °C until DNA extraction. Soil physical and chemical properties were taken from 4 cm diameter by 15 cm deep cores co-located with the high-throughput sequencing cores. These included total C (%), N (%), P (mg kg$^{-1}$), organic matter (% LOI), pH (measured in 0.01 M CaCl$_2$), mean soil water repellency (median water drop penetration time in seconds), bulk density (g cm$^{3}$ $^{-1}$), volume of rocks (cm$^3$), soil bound water (g water g dry soil$^{-1}$), volumetric water content (m$^3$ m$^{-3}$ $^{-1}$), as well as clay and sand content (%) of soil. Abundances of mesofauna collected as part of GMEP were taken from George et al.[23] and geographic data including grid eastings, northings, and elevation were also included in our analyses. For complete details on chemical analyses, see Emmett et al.[51]. Temperature (°C) and mean annual precipitation (mL) were extracted from the Climate Hydrology and Ecology research Support System dataset[54]. Mean values for each variable were recorded for each AVC (Supplementary Table 12) and soil properties were normalised where appropriate.

Soil texture data were measured by laser granulometry with a LS320 13 analyser (Beckman-Coulter). We subsampled approximately 0.5 g of soil taken from 15 cm cores by manual quartering and removed organic C using H$_2$O$_2$ and then transferred the sample into 250 mL bottles, added 5 mL of 5% Calgon® and shook overnight at 240 rpm. Bottles were emptied manually into the laser diffraction instrument for measuring particle size distribution. Full Mie theory was used to obtain a particle size distribution from the raw measurement data, with the real refractive index set to 1.55 and the absorption coefficient at 0.1 as in Özer et al.[55]. The cut-off points for clay, silt, and sand were 2.2, 63, and 2000 µm, respectively. Clay and sand percentages were selected for subsequent analyses and normalised using Aitchison's log-ratio transformation.

**DNA extraction**. Soils were homogenised by passing through a sterilised 2 mm stainless steel sieve. Sieves were sterilised between samples by rinsing under the tap water using high flow, applying Vircon laboratory disinfectant and UV-treating each side for 5 min DNA was extracted by mechanical lysis and the homogenisation step performed in triplicate from 0.25 g of soil per sample using a PowerLyzer PowerSoil DNA Isolation Kit (MO-BIO). Pre-treatment with 750 µL of 1 M CaCO$_3$ following Sagova-Mareckova et al.[56] was performed as it was shown to improve PCR performances, especially for acidic soils. Extracted DNA was stored at −20 °C until amplicon library preparation began. To check for contamination in sieves 3 negative control DNA extractions were completed and an additional 2 negative control kit extractions were performed using the same technique but without the CaCO$_3$ solution.

**Primer selection and PCR protocols for library preparation**. Amplicon libraries were created using primers for rRNA marker genes, specifically for the V4 region of the 16S rDNA gene targeting bacteria and archaea (515F/806R)[57], ITS1 targeting fungi (ITS5/5.8S_fungi)[58], and the V4 region of the 18S rDNA gene (TAReuk454FWD1/TAReukREV3)[59] targeting a wide range of, but not all, eukaryotic organisms. We used a two-step PCR following protocols devised in conjunction with the Liverpool Centre for Genome Research. Amplification of amplicon libraries was run in triplicate on DNA Engine Tetrad® 2 Peltier Thermal Cycler (BIO-RAD Laboratories) and thermocycling parameters for each PCR started with 98 °C for 30 s and terminated with 72 °C for 10 min for final extension and held at 4 °C for a final 10 min For the 16S locus, first-round PCR amplification followed 10 cycles of 98 °C for 10 s; 50 °C for 30 s; 72 °C for 30 s. For ITS1, there were 15 cycles of 98 °C for 10 s; 58 °C for 30 s; 72 °C for 30 s. For 18S there were 15 cycles at 98 °C for 10 s; 50 °C for 30 s; 72 °C for 30 s. Twelve µL of each first-round PCR product was mixed with 0.1 µL of exonuclease I, 0.2 µL thermosensitive alkaline phosphatase, and 0.7 µL of water and cleaned in the thermocycler with a programme of 37 °C for 15 min and 74 °C for 15 min and held at 4 °C. Addition of Illumina Nextera XT 384-way indexing primers to the cleaned first-round PCR products were amplified following a single protocol which started with initial denaturation at 98 °C for 3 min; 15 cycles of 95 °C for 30 s; 55 °C for 30 s; 72 °C for 30 s; final extension at 72 °C for 5 min and held at 4 °C. Twenty-five µL of second-round PCR products were purified with an equal amount of AMPure XP beads (Beckman Coulter). Library preparation for 2013 samples was conducted at Bangor University. Illumina sequencing for both years and library preparation for 2014 samples were conducted at the Liverpool Centre for Genome Research.

**Bioinformatics**. Bioinformatics analyses were performed on the Supercomputing Wales cluster. A total of 130,219,260, 104,276,828, and 98,999,009 raw reads were recovered from the 16S, ITS1, and 18S sequences, respectively. Illumina adapters were trimmed from sequences using Cutadapt[60] with 10% level mismatch for removal. Sequences were then de-multiplexed, filtered, quality-checked, and clustered using a combination of USEARCH v. 7.0[61] and VSEARCH v. 2.3.2[62]. Open-reference clustering (97% sequence similarity) of OTUs was performed using VSEARCH; all other steps were conducted with USEARCH. Sequences with a maximum error greater than 1 and shorter than 200 bp were removed following the merging of forward and reverse reads for 16S and ITS1 sequences. A cut-off of 250 bp was used for 18S sequences, according to higher quality scores. There were 15,202,313 (16S), 7,242,508 (ITS1), and 9,163,754 (18S) cleaned reads left at the end of these steps. Sequences were sorted and those that only appeared once in the dataset were removed. Briefly, filtered sequences were matched first against a number of different reference databases: Greengenes 13.8[63], UNITE 7.2[64], and SILVA 128[65] for 16S, ITS1, 18S, respectively. Ten percent of sequences that failed to match were clustered de novo and used as a new reference database for failed sequences. Sequences that failed to match with the de novo database were subsequently also clustered de novo. All clusters were collated and chimeras were removed using the uchime_ref command in VSEARCH.

Chimera-free clusters and taxonomy assignment were used to create an OTU table with QIIME v. 1.9.1[66] using RDP[67] methodology with the GreenGenes database for 16S and UNITE database for ITS1 data. Taxonomy was assigned to the 18S OTU table using BLAST[68] against the SILVA database and OTUs appearing only once or in only 1 sample were removed from each OTU table.

Newick trees were constructed for the 16S and 18S tables using 80% identity thresholds. The trees were combined with their respective OTU tables as part of analyses using the R package phyloseq[69], removing OTUs that did not appear in both the tree and OTU table. OTUs identified as eukaryotes in the 16S OTU table, non-fungi OTUs in the ITS OTU table, as well as OTUs identified as fungi, plants, and non-soil animals were removed from the 18S OTU table. Read counts from each group were normalised using rarefaction. The OTU tables were rarefied 100 times using phyloseq[69] (as justified by Weiss et al.[70]) and the resulting mean richness was calculated for each sample. The read depth used for rarefaction varied for each group (Supplementary Table 14). Samples with lower read counts than this cut-off were removed before rarefaction. A summary of number of replicates per AVC is included in Supplementary Table 1.

**Statistical analyses**. All statistical analyses were run using R v. 3.3.3[71] using the rarefied data sets for each organismal group. The vegan package[72] was used to assess β-diversity via NMDS and CAP ordinations based on Bray–Curtis dissimilarities. A linear model for each environmental variable was fit separately to the ordination using the envfit function, the results are presented ranked according to goodness-of-fit. Results of goodness-of-fit for each variable from both ordination methods were compared using regression analyses to look for congruence. The values of all variables were plotted against NMDS scores to determine if there were positive or negative relationships with each NMDS axis. Differences in β-diversity amongst AVCs were calculated with PERMANOVA. The assumption of homogeneity of dispersion was verified using the betadisper function.

Linear mixed models were constructed using package nlme[73] to test the differences in α-diversity amongst AVCs for each organismal group. Model selection was performed using AVC, soil type, LOI classification, and sample year as fixed factors; sample square identity was the random factor. To determine the best possible model, predictors other than AVC were dropped to find the lowest AIC scores using the AICcmodavg package[74]. For each model, significant differences were assessed by ANOVA and pairwise differences were identified with Tukey's post-hoc tests from the multcomp package[75].

PLS regressions found in package pls[76] were used to identify the most important environmental variables for richness. Such analysis is ideal for data where there are many more explanatory variables than sample numbers or where extreme multicollinearity is present[46]. As in Lallias et al.[46], we used the variable importance in projection (VIP) approach[77] to sort the original explanatory variables by order of importance; variables with VIP values > 1 were considered most important. Relationships between important variables and richness values for each group of organisms were investigated by linear regression. Richness was normalised before regression when necessary. Pearson's correlation coefficient was used to directly compare richness of organismal groups.

**Reporting summary**. Further information on experimental design is available in the Nature Research Reporting Summary linked to this article.

## Data availability

Data associated with this paper will be publically published in the National Environment Research Council (NERC) Environmental Information Data Centre (EIDC). Sequences with limited sample metadata have been uploaded to The European Nucleotide Archive and can be accessed with the following primary accession codes after the end of data embargo (27 June, 2020): PRJEB27883 (16S), PRJEB28028 (ITS), and PRJEB28067 (18S). Data are also available from the authors upon reasonable request with permission from the Welsh Government. The source data underlying Fig. 3a–e is provided as a Source Data file.

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

## Acknowledgements

This research was supported under the Glastir Monitoring & Evaluation Programme (Contract reference: C147/2010/11) and by the UK Natural Environment Research Council (NERC) through the RC/Centre for Ecology & Hydrology (CEH Project: NEC04782). P.B.L.G. was supported by a Soils Training and Research Studentship (STARS) grant from the Biotechnology and Biological Sciences Research Council and NERC [Grant number NE/M009106/1]. STARS is a consortium consisting of Bangor University, British Geological Survey, Centre for Ecology and Hydrology, Cranfield University, James Hutton Institute, Lancaster University, Rothamsted Research, and the University of Nottingham. The authors thank the GMEP team for their contribution in collecting the data, especially Aidan Keith, for assistance in enumerating mesofauna, and the laboratory staff of Environment Centre Wales. The authors appreciate the support

and assistance of Supercomputing Wales for bioinformatics analyses. Sequence data generation was performed at the Centre for Genomic Research, University of Liverpool.

## Author contributions

D.L.J., D.A.R., S.C., and B.A.E. conceived this project. D.L., D.L.J., and I.L. processed the soil samples and collected data with quality assurance by F.M.S. D.L. and S.C. led the DNA extraction, primer design, library generation, and established the bioinformatics pipelines. J.G.K. and R.M.E. led sequence data generation and assisted with library preparations. Bioinformatics and statistical analyses were led by P.B.L.G. with assistance from S.C., R.I.G., D.L., and F.M.S. P.B.L.G. wrote the first draft of the manuscript and F.M.S., S.C., D.L., R.I.G., D.A.R., and D.L.J. contributed to subsequent revisions. All authors read and approved the final draft of the manuscript.

## Additional information

**Competing interests:** The authors declare no competing interests.

