## [Peer Review File · Nature Communications]

This manuscript has been previously reviewed at another journal that is not operating a transparent peer review scheme. This document only contains reviewer comments and rebuttal letters for versions considered at Nature Communications.

Reviewers' Comments:

Reviewer #2:

Remarks to the Author:

The MS sets out to be rather challenging a number of assumptions in soil ecology but is a bit sweeping in its statements on the novelty. There is a wealth of publications and work on this matter, and this MS is not the first testing how different organisms in soil are related to chemical physical factors or to land use. The authors claim that they have changed the text since the last submission, I do find more references, but they still make conclusions and aims that this manuscript claims to "challenge existing ecologic paradigms regarding how soil organisms respond to soil factors and land use" (see line 36, 64, 95). The main findings that bacteria and fungi can be correlated to soil pH and microarthropods to land use is not novel and is the standard knowledge of today. The work on these matters has been extensively published for the last 40 years. The biogeography of soil microorganisms have been described in a number of papers over the last 10 years, with the novel sequencing techniques. Thus the results are not challenging ecological paradigms, as this is rather mainstream knowledge of today.

They have done an ambitious sampling scheme and sequenced hundreds of samples for different organisms, though the scope of the paper should have used a larger geographical areas to test some of the assumptions that they make. The paper does not represent diversity across temperate soil ecosystems, it still represents different ecosystems in Wales. In the introduction line 67-87, describes general patterns of diversity of soil organisms across latitudinal gradients on continental and global scales, though this study has a more limited scope.

The gradient of sample sites are based on vegetation classes, and then it is stated that the AVC are linked to line 107 soil nutrient conditions from which soil productivity and management intensity can be inferred. However this is based on a non scientifically published report, thus not reliable for a base of a scientific publication (Bunce et al 1990) as it is not possible to evaluate the way this classification of intensity has been done. So there is still no reliable explanation of the intensity gradient and the links to the AVC. They have measured several soil factors so I don't understand why these are not the base of the nutrient gradient and the intensity gradient can be based on other scientific publications working on gradient of management intensity.

In the presentation, the outline of the results is rather based on the methods and not on the scope of the paper, It is not interesting to know how many sequences they have found in total. In a general paper like this concerning the diversity patterns, this is what should be the driver of the story , not the sequencing results and the number of OTUs from that. So the main part of the text under sequencing should rather be a part of the appendix or the methods. Further they also fall into the trap to let the statistic lead the story of the text instead of letting the significant biological results be the main story (line 145)

When regarding the results and how this have been determined, the sequencing seems to be done according to current methods and with appropriate bioinformatics. When evaluating what the results actually means It is not clear how they have worked out the average numbers and total abundances of species per vegetation class as there are different numbers of samples within each class. In Fig 1 and 2 it seems to be the total abundance of OTUs, but are they normalized in any way to cover the differences between sampling effort? Also in Fig 3, what is the base of the average value? When you have a larger sampling effort, there could be more species, and this could influence the results more than the land use drivers.

The use of NMDS and correlations to the soil factors is still not done or explained satisfactory. Now the

authors are more vague in how they described the method that was used to what was submitted initially, see lines 465-468. It is not clear to me whether the environmental variables actually have been included in the NMDS or that the scores from the NMDS have been used. Now the result shows the regression coefficients of each environmental variable explaining the composition of communities in two dimensional ordination space. This is somewhat questionable, as you do for example not account for the co-linearity of environmental variables in an NMDS. In a PCA you would plot all the environmental variables as vectors and would get a visual representation of their relationship to the axis scores and each other. NMDS ordination space is Euclidean, but the resemblance measure is not (as here Bray-Curtis). It is not generally recommended to extract correlations between environmental variables and axis scores from NMDS (by Oksanen: <https://stat.ethz.ch/pipermail/r-sig-ecology/2010-August/001448.html>). Instead one could use distance-based linear models to model these relationships (vegan command: `capscale`). So in order to resolve this, the authors need to explain in much more detail what they have done and how they have done the analyses, and how they themselves can explain what they have done.

Also did you use abundances in the NMDS or normalized values or presence absence, I can't find information on this. This is important as you are discussing diversity effects and some ordination techniques may show the differences more due to abundances than to community differences as the number in individuals will differ in the different samples.

You still state that you use abundances of microarthropods as soil factors when correlating to the sequencing results, which should not be done as they are dependent on each other.

Reviewer #4:

Remarks to the Author:

The authors found that animal diversity follow a different trend than microbial diversity in temperate ecosystem across Wales. This is an interesting case of study, however, I am not sure whether the results included here are novel and representative enough to be published in this high-profile journal. The fact that soil communities are indirectly regulated by land use via shifting soil properties is well-known. Also, the diversity of soil animals is known to be impacted by cropping. Moreover, we know that bacteria are less sensitive to land use intensification than soil fauna in temperate ecosystems from Europe (Gossner et al. 2016, Nature), leading to different spatial patterns for these taxa. More importantly, there are ecological theories supporting the idea -already proven wrong- that the diversity of different taxa should follow similar latitudinal and elevational patterns at the global scale-, however, a strong theory supporting why the diversity of different soil taxa should follow similar patterns at the scale of Wales is lacking.

Line 1. Add "from Wales".

Lines 30. Not sure why a correlation between diversity of different groups of animals and microbes is expected at the national scale? Is there any strong theory behind this assumption?

Lines 30-31. This is misleading, as samples were collected at the scale of Wales, which is not representative for many temperate ecosystems on Earth.

Lines 33-35. Therefore, land use was the major driver of microbial diversity via changing soil properties?

Lines 35-38. Not sure about what paradigm the authors are challenging here.

Line 49. Amplicon sequencing has been around for a while now.

Lines 62-65. We are far past these early stages. See Fierer (2017; Nature Reviews Microbiology), Ramirez et al. (2014; Proceedings of the Royal Society B: Biological Sciences) and Tedersoo et al. (2014; Science).

Lines 64-65. What paradigm is without a proper validation?

Lines 68-69. The opposite has been also reported. See Siles and Margesin (2016) *Microbial Ecology*.

Line 71. See Bahram et al. (2018). *Nature*. Bacterial diversity peaks at mid-latitudes globally.

Line 76. This is only the case for bacteria.

Lines 81-84. These lines do not reflect the current level of knowledge on this topic.

Lines 96-98. Aren't these soil properties driven by land use as well?

Lines 145-151. Was climate and location considered in these analyses? Bogs are often in colder and more mesic locations than croplands in the UK. Do this have to do with productivity? Bogs often have oxygen limitations, low pH, very high C:N ratio and lots of C, while croplands have completely opposite soil conditions.

Lines 151-153. How was the correlation between diversity and richness of fungi, protists, bacteria, archaea and animals? Considering the topic of your study, the reader might wonder about this essential information.

Line 169. You might consider running some path analyses to identify the direct and indirect effects of land use on soil biodiversity. At the moment, the discussion on potential indirect effects of land use on microbial communities via changes in soil properties is a bit informal.

Lines 214-215. Isn't this the expected?

Reviewer #5:

Remarks to the Author:

The present manuscript by George et al. reports on a comprehensive soil sequencing study of bacteria, fungi, archaea, protists, and animals across >400 locations in Wales. The authors find that the diversity of soil microbes and animals have dissimilar drivers, with microbes being mostly affected by soil conditions and soil animals responding to land use intensity. These divergent patterns based on a very comprehensive dataset are highly interesting to a broad scientific audience. It was a great pleasure to read this manuscript, and, as far as I can judge, the methods related to sampling and analyses are appropriate. Although I was not involved in the first review round, I share the positive view of the previous reviewers and think that the reviewers' comments were addressed in a satisfactory way. I am aware of the fact that I may now come with a completely novel set of concerns; however, I do think they should be addressable with a minor revision and hopefully help to further improve the paper.

First, I think that the headers in the Results section should not be named according to methods, but to main drivers or response variables. Second, as written, the section of the Discussion between lines 219 and 225 question the novelty of the paper. It should be made more clear what the novel finding of this study is in comparison to others, such as the ones conducted in France. Third, although I appreciate the attempt to apply general ecological principles to the observed patterns, I am not sure if this can be done without any caveats statements. For example, the intermediate disturbance hypothesis is mentioned as one potential mechanism why disturbed arable lands harbor high microbial diversity. However, I am wondering if successional dynamics that are required to contribute to the intermediate disturbance hypothesis function at the appropriate time scale? Bacteria produce multiple generations per day, and disturbances in arable fields may rather affect processes at evolutionary time scales than at successional time scales. Some more thoughts on this potential issue could be helpful. Forth, the current manuscript touches very little on the different spatial scales that affect the different target organisms. Soil cores may integrate multiple habitats for soil mesofauna, but different "continents" for microbes. How will these different spatial (and also temporal scales; see point #3) affect the conclusions, such as in lines 348-352? Last and very minor suggestions: I suggest to add the taxa names to the different panels. Moreover, it would be very helpful to have information on the level of replication per habitat type in Figures 2 and 3.

Reviewer #2 (Remarks to the Author):

The MS sets out to be rather challenging a number of assumptions in soil ecology but is a bit sweeping in its statements on the novelty. There is a wealth of publications and work on this matter, and this MS is not the first testing how different organisms in soil are related to chemical physical factors or to land use. The authors claim that they have changed the text since the last submission, I do find more references, but they still make conclusions and aims that this manuscript claims to “challenge existing ecologic paradigms regarding how soil organisms respond to soil factors and land use” (see line 36, 64, 95). The main findings that bacteria and fungi can be correlated to soil pH and microarthropods to land use is not novel and is the standard knowledge of today. The work on these matters has been extensively published for the last 40 years. The biogeography of soil microorganisms have been described in a number of papers over the last 10 years, with the novel sequencing techniques. Thus the results are not challenging ecological paradigms, as this is rather mainstream knowledge of today.

1. We have rewritten both the introduction (lines 25-38) and abstract (lines 57-91) to more effectively convey the aims and objectives of our study. We have removed the sweeping statements about challenging ecological paradigms and have focused our text to make it clear that we are testing divergent responses to land use and soil properties in the microbial and animal fractions of soil communities and whether these persist across heterogeneous systems at the national-level using a standardised metabarcoding approach. We have used extensive work of the past 10-15 years to contextualise our research. Further, we are aware of the work to date, but the novelty of our synthesis is the standardization of approaches to address the key questions across broad habitat types that are not confounded by space, time or ecology.

They have done an ambitious sampling scheme and sequenced hundreds of samples for different organisms, though the scope of the paper should have used a larger geographical areas to test some of the assumptions that they make. The paper does not represent diversity across temperate soil ecosystems, it still represents different ecosystems in Wales.

2. We have further defined our study system within the title to highlight how our work encompasses a diversity of natural systems within a geographic range that limits effects of confounding ecological variables.

In the introduction line 67-87, describes general patterns of diversity of soil organisms across latitudinal gradients on continental and global scales, though this study has a more limited scope.

3. As stated earlier (Response 1), we have revised our introduction to make the aims of our study clearer. We have retained the description of some of these large-scale patterns as they both showcase the current state of the literature. However, we have tried to better focus our aims in this section (lines 57-84). Although our study has a

defined geographic scope, the area necessarily encompasses a diverse range of land uses and ecosystems. Furthermore, since these heterogeneous landscapes are in a relatively constrained area, confounding effects from local climate and land use history are minimised. This has allowed us greater confidence that our findings are reflective of large-scale diversity patterns as opposed to those found within a singular ecosystem type (i.e. grasslands in Gossner et al., 2016).

The gradient of sample sites are based on vegetation classes, and then it is stated that the AVC are linked to line 107 soil nutrient conditions from which soil productivity and management intensity can be inferred. However this is based on a non scientifically published report, thus not reliable for a base of a scientific publication (Bunce et al 1990) as it is not possible to evaluate the way this classification of intensity has been done. So there is still no reliable explanation of the intensity gradient and the links to the AVC. They have measured several soil factors so I don't understand why these are not the base of the nutrient gradient and the intensity gradient can be based on other scientific publications working on gradient of management intensity.

4. With respect, we disagree with this dismissal of Bunce et al. (1999). The AVC system captures the land use and belowground properties by using the specific plants present in the sampling area. Therefore it is more directly linked to land use than simple habitat assessments. Papers including Smart et al. (2003), Maskell et al. (2013), and George et al. (2017) have used the AVC productivity/intensity gradient and generated a combined 101 citations. Furthermore, this classification system is an integral part of British national soil and vegetation surveys (e.g. Emmett et al., 2008; Emmett et al. 2015) and has been used extensively in related publications (Black et al., 2003; Griffiths et al., 2011).

In the presentation, the outline of the results is rather based on the methods and not on the scope of the paper, It is not interesting to know how many sequences they have found in total. In a general paper like this concerning the diversity patterns, this is what should be the driver of the story, not the sequencing results and the number of OTUS from that. So the main part of the text under sequencing should rather be a part of the appendix or the methods.

5. We respectfully disagree with this criticism of our presentation. Leading the results section with sequencing results is a common practice in similar papers in Nature-group journals (Bista et al., 2017; Mahé et al., 2017). We therefore prefer to retain this section if possible.

Further they also fall into the trap to let the statistic lead the story of the text instead of letting the significant biological results be the main story (line 145)

6. We agree and have amended this line (see line 152).

When regarding the results and how this have been determined, the sequencing

seems to be done according to current methods and with appropriate bioinformatics. When evaluating what the results actually means It is not clear how they have worked out the average numbers and total abundances of species per vegetation class as there are different numbers of samples within each class. In Fig 1 and 2 it seems to be the total abundance of OTUs, but are they normalized in any way to cover the differences between sampling effort?

7. The OTU tables were normalised through rarefaction. This process standardises the number of reads across all samples. In order to account for errors introduced by random starts etc. OTU tables were rarefied 100 times, and averaged to obtain richness values. This means that there will be the same amount of reads in each sample, regardless of replicate number across land uses. All figures are based on normalised data (line 485-486 added) and the proportional abundances presented in Fig. 2 were calculated individually within each AVC.

Also in Fig 3, what is the base of the average value? When you have a larger sampling effort, there could be more species, and this could influence the results more than the land use drivers.

8. As our data is derived from a large-scale soil survey, we were unable to standardise the number of replicates per each land use category. However, by standardising the number of reads in each sample to a set depth, we overcome the potential issues presented by uneven sample distributions within land uses. We have also repeated the analysis with an even number of replicates per AVC ($n = 9$) and the observed trends are still apparent at this level.

The use of NMDS and correlations to the soil factors is still not done or explained satisfactory. Now the authors are more vague in how they described the method that was used to what was submitted initially, see lines 465-468. It is not clear to me whether the environmental variables actually have been included in the NMDS or that the scores from the NMDS have been used. Now the result shows the regression coefficients of each environmental variable explaining the composition of communities in two dimensional ordination space. This is somewhat questionable, as you do for example not account for the co-linearity of environmental variables in an NMDS. In a PCA you would plot all the environmental variables as vectors and would get a visual representation of their relationship to the axis scores and each other. NMDS ordination space is Euclidean, but the resemblance measure is not (as here Bray-Curtis). It is not generally recommended to extract correlations between environmental variables and axis scores from NMDS (by Oksanen: <https://stat.ethz.ch/pipermail/r-sig-ecology/2010-August/001448.html>). Instead one could use distance-based linear models to model these relationships (vegan command: capscale). So in order to resolve this, the authors need to explain in much more detail what they have done and how they have done the analyses, and how they themselves can explain what they have done.

10. We feel the reviewer has misunderstood the intention of our analyses here, and so have edited the text to make this clearer. The methodology of fitting environmental vectors to ordination sample site scores (`vegan::envfit`) is commonplace in ecology (Griffiths et al., 2011; Read et al., 2014; Chimienti et al., 2016) for assessing relationships between discrete environmental variables and the x+y location of a sample in two dimensional ordination space. Co linearity is not an issue here because each environmental vector is fitted to the ordination site scores separately. The criticism in relation to the mailing list we think refers to an issue that NMDS ordination axes (1 or 2) should not be used in alone in a statistical analyses. We have not done this here, and we have further explained our methodology in lines 486-493.

We have however re-analysed the ordination data using the CAP constrained-ordination technique recommended by Reviewer 2. Here we show the CAP ordination of the bacterial community on a subset of environmental variables (Fig. 1). In addition, we present the results of linear fitting to the NMDS ordination in the same presentation style (Fig. 2). These figures show the same results, however we feel that these figures are cluttered by the sheer amount of variables that were studied. We therefore believe that our original presentation of this data in table form is more appropriate.

The goodness-of-fit for each vector from the `envfit` function was largely similar across both ordinations; however, climatic variables were given higher R^2 values under CAP ordination than NMDS (Table 1 and 2). This is likely because we are interested in is so many variables that a constrained ordination method, i.e. CAP, will not differ much from an unconstrained ordination, i.e. NMDS (Fig. 1-2; <http://cc.oulu.fi/~jarioksa/opetus/metodi/sessio2.pdf>).

However, we would be happy to amend the manuscript to include the CAP ordination results if there was a strong editorial push to do so.

Fig. 1. Plot of the CAP ordination of bacterial community composition across GMEP sites. Samples are coloured by Aggregate Vegetation Class. Results of PERMANOVA ($F_{6,341} = 17.18$, $p = 0.001$) and dispersion of variances of groups ($F_{6,335} = 6.48$, $p = 0.001$) were significant. Arrows represent the linearly fitted environmental variables

Fig. 2. Plot of the non-metric dimensional scaling ordination (stress = 0.06) of bacterial community composition across GMEP sites. Samples are coloured by Aggregate Vegetation Class. Results of PERMANOVA ($F_{6,427} = 30.76$, $p = 0.001$) and dispersion of variances of groups ($F_{6,427} = 10.97$, $p = 0.001$) were significant.

Table 1. Summary of relationships amongst environmental factors and bacteria communities.

Soil and environmental variables	R ²
pH	0.66***
Mean annual precipitation (mL)	0.51***
C :N ratio ^S	0.48***
Elevation (m)	0.47***
Volumetric water content (m ³ /m ³)	0.46***
Bulk density (g/cm ³)	0.44***
Organic matter (% LOI) ^L	0.39***
Total C ^L	0.32***
Clay content (%) ^A	0.29***
Soil bound water (g water per g of dry soil)	0.26***
Soil water repellency ^L	0.26***
Sand content (%) ^A	0.22***
Total N (%) ^L	0.22***
Total P (mg/kg) ^S	0.09***
Collembola ^{L1}	0.09***
Total mesofauna ^{L1}	0.08***
Mites ^{L1}	0.08***
Rock volume (mL)	0.05***
Temperature (°C)	0.04**

Note: ^A denotes Aitchison's log-ratio transformation; ^L denotes log₁₀-transformation; ^{L1} denotes log₁₀ plus 1 transformation ^S denotes square-root-transformation

Table 2. Summary of relationships amongst environmental factors and bacteria communities. +/- signify the direction of association between each variable and respective NMDS axes.

Soil and environmental variables	R ²	Correlation		
		Axis1	Axis2	Axis3
pH	0.71***	-	-	+
C :N ratio ^S	0.52***	+	-	+
Volumetric water content (m ³ /m ³)	0.49***	+	-	+
Bulk density (g/cm ³)	0.47***	-	+	-
Organic matter (% LOI) ^L	0.46***	+	-	+
Elevation (m)	0.45***	+	-	-
Mean annual precipitation (mL)	0.43***	+	-	-
Total C ^L	0.39***	+	-	+
Clay content (%) ^A	0.33***	-	+	-
Soil bound water (g water per g of dry soil)	0.31***	+	-	+
Soil water repellency ^L	0.27***	+	-	-
Total N (%) ^L	0.26***	+	-	+
Sand content (%) ^A	0.21***	+	+	+
Collembola ^{L1}	0.09***	-	+	-
Mites ^{L1}	0.06***	+	+	-
Total P (mg/kg) ^S	0.06***	-	-	-
Total mesofauna ^{L1}	0.06***	+	+	-
Rock volume (mL)	0.05**	-	+	-
Temperature (°C)	0.03*	+	+	-

Note: ^A denotes Aitchison's log-ratio transformation; ^L denotes log₁₀-transformation; ^{L1} denotes log₁₀ plus 1 transformation ^S denotes square-root-transformation

Also did you use abundances in the NMDS or normalized values or presence absence, I can't find information on this. This is important as you are discussing diversity effects and some ordination techniques may show the differences more due to abundances than to community differences as the number in individuals will differ in the different samples.

11. To clarify, the normalised (rarefied) OTU tables were used in NMDS ordinations, and for all other statistics. A statement to this effect has been added in line 485-486.

You still state that you use abundances of microarthropods as soil factors when correlating to the sequencing results, which should not be done as they are dependent on each other.

12. Again, we think this is important to demonstrate that the two independent data sets (mesofauna collected and enumerated by traditionally taxonomy vs eDNA signals) are showing similar patterns. If they did not correlate there would be serious issues

with connecting DNA and morphological data sets and, in our opinion, such a connection provides added depth to our data synthesis

Reviewer #4 (Remarks to the Author):

The authors found that animal diversity follow a different trend than microbial diversity in temperate ecosystem across Wales. This is an interesting case of study, however, I am not sure whether the results included here are novel and representative enough to be published in this high-profile journal. The fact that soil communities are indirectly regulated by land use via shifting soil properties is well-known. Also, the diversity of soil animals is known to be impacted by cropping. Moreover, we know that bacteria are less sensitive to land use intensification than soil fauna in temperate ecosystems from Europe (Gossner et al. 2016, Nature), leading to different spatial patterns for these taxa. More importantly, there are ecological theories supporting the idea -already proven wrong- that the diversity of different taxa should follow similar latitudinal and elevational patterns at the global scale-, however, a strong theory supporting why the diversity of different soil taxa should follow similar patterns at the scale of Wales is lacking.

13. As stated in Response 1, we have now clearly stated our specific aims and objectives and removed the broader statements (see lines 85-91). Specifically, we have focused our text to make it clear that we are testing whether divergent responses to land use and soil properties in the microbial and animal fractions of soil communities persist across heterogeneous systems at the national-level using a standardised metabarcoding approach.

Line 1. Add “from Wales”.

14. We feel that the revised title adequately reflects the study area and are mindful that localisation of science in titles can often lead to reduced citations and engagement in the work that we are particularly passionate about.

Lines 30. Not sure why a correlation between diversity of different groups of animals and microbes is expected at the national scale? Is there any strong theory behind this assumption?

15. Previous studies demonstrating the relative resistance of bacteria and sensitivity of soil fauna to land use intensification have focused on relatively homogeneous grasslands and involve mixed sampling schemes (i.e. grasslands with mixed metabarcoding and taxonomic assessments in Gossner et al., 2016). We believe that the strength of our study is in the fact that sampling across Wales has allowed us to test if these findings extend across a wider area with very heterogeneous landscapes (see Response 1). Although we may have struggled to convey this adequately in previous drafts, we feel that our summation of our aims in lines 28-31 now conveys this clearly to the reader.

Lines 30-31. This is misleading, as samples were collected at the scale of Wales, which is not representative for many temperate ecosystems on Earth.

16. Yes, point taken. This has been addressed by further defining our sampling area as national-scale analysis of 7 temperate systems.

Lines 33-35. Therefore, land use was the major driver of microbial diversity via changing soil properties?

17. Yes, land use is a key driver of microbial communities. While it is clear that vegetation can alter soil properties, it is clear that this is bi-directional.

Lines 35-38. Not sure about what paradigm the authors are challenging here.

18. These lines have been revised to reflect the aims of our study more clearly. As stated previously (Response 1), the objective of our study was to determine if divergent responses to land use and soil properties in the microbial and animal fractions of soil communities persist across heterogeneous systems at the national-level using a standardised metabarcoding approach (line 28-31).

Line 49. Amplicon sequencing has been around for a while now.

19. Agreed. This line has been removed.

Lines 62-65. We are far past these early stages. See Fierer (2017; Nature Reviews Microbiology), Ramirez et al. (2014; Proceedings of the Royal Society B: Biological Sciences) and Tedersoo et al. (2014; Science).

20. We have appropriately acknowledged the extent of the advancements in the field in recent years (lines 60-84).

Lines 64-65. What paradigm is without a proper validation?

21. This line has been removed

Lines 68-69. The opposite has been also reported. See Siles and Margesin (2016) Microbial Ecology.

Line 71. See Bahram et al. (2018). Nature. Bacterial diversity peaks at mid-latitudes globally.

22. We thank Reviewer 4 for these informative and important papers. However, in our efforts to better define the scope and aims of our manuscript we have removed the text to which they have been recommended.

Line 76. This is only the case for bacteria.

23. We respectfully disagree. Terdersoo et al (2014) found that pH is the most important soil property for some – i.e. mycorrhizal – but not all fungi. In addition, research from the UK has shown the dominant influence of pH on protists (Dupont et al., 2016) and mesofauna (George et al., 2017). Therefore we believe that this revised statement about the dominant role pH in shaping belowground communities is justified (line 78-79).

Lines 81-84. These lines do not reflect the current level of knowledge on this topic.

24. These lines have been removed.

Lines 96-98. Aren't these soil properties driven by land use as well?

25. Please see Response 17.

Lines 145-151. Was climate and location considered in these analyses? Bogs are often in colder and more mesic locations than croplands in the UK. Do this have to do with productivity? Bogs often have oxygen limitations, low pH, very high C:N ratio and lots of C, while croplands have completely opposite soil conditions.

26. We are unsure as to what Reviewer 4 is asking here. The climatic and geographic aspects of bogs are inherent in the AVC classification scheme (as explained in the Supplementary Appendix), along with productivity. The low pH, high C:N ratio etc. are accounted for within the classification scheme and can be seen in our data on soil properties (Supplementary Table 7).

Lines 151-153. How was the correlation between diversity and richness of fungi, protists, bacteria, archaea and animals? Considering the topic of your study, the reader might wonder about this essential information.

27. As requested, we have included correlations of richness between each group in Supplementary Table 2. Lines 176-184 direct the reader to this new information.

Line 169. You might consider running some path analyses to identify the direct and indirect effects of land use on soil biodiversity. At the moment, the discussion on potential indirect effects of land use on microbial communities via changes in soil properties is a bit informal.

28. This is a good point. We have now run preliminary structural equation models investigating this aspect of the project. However, they have not proven informative, as they only highlight the importance of pH in structuring microbial richness. Furthermore they do not have sufficiently strong R^2 values for us to be confident in publishing their findings in this manuscript. Therefore we do not feel that their inclusion in this manuscript is appropriate at the present time.

Lines 214-215. Isn't this the expected?

29. Although these results have been presented previously (i.e. Gossner et al., 2016), we feel that these findings have been quite limited in scope. By showing that such divergent responses to land use in microbes and fauna extend across heterogeneous landscapes, we believe our results provide a necessary addition to the framework of soil communities at the higher levels.

Reviewer #5 (Remarks to the Author):

The present manuscript by George et al. reports on a comprehensive soil sequencing study of bacteria, fungi, archaea, protists, and animals across >400 locations in Wales. The authors find that the diversity of soil microbes and animals have dissimilar drivers, with microbes being mostly affected by soil conditions and soil animals responding to land use intensity. These divergent patterns based on a very comprehensive dataset are highly interesting to a broad scientific audience. It was a great pleasure to read this manuscript, and, as far as I can judge, the methods related to sampling and analyses are appropriate. Although I was not involved in the first review round, I share the positive view of the previous reviewers and think that the reviewers' comments were addressed in a satisfactory way. I am aware of the fact that I may now come with a completely novel set of concerns; however, I do think they should be addressable with a minor revision and hopefully help to further improve the paper.

First, I think that the headers in the Results section should not be named according to methods, but to main drivers or response variables.

30. We have renamed the results subsections with more informative titles.

Second, as written, the section of the Discussion between lines 219 and 225 question the novelty of the paper. It should be made more clear what the novel finding of this study is in comparison to others, such as the ones conducted in France.

31. We thank Reviewer 5 for highlighting this shortcoming. We have added lines 236-245 to better showcase the novelty of our findings.

Third, although I appreciate the attempt to apply general ecological principles to the observed patterns, I am not sure if this can be done without any caveats statements. For example, the intermediate disturbance hypothesis is mentioned as one potential mechanism why disturbed arable lands harbor high microbial diversity. However, I am wondering if successional dynamics that are required to contribute to the intermediate disturbance hypothesis function at the appropriate time scale? Bacteria produce multiple generations per day, and disturbances in arable fields may rather affect processes at evolutionary time scales than at successional time scales. Some more thoughts on this potential issue could be helpful.

32. Reviewer 5 has raised a good point here. We have added lines 257-259. This passage highlights the fact that microbes, bacteria and protists especially, often enter dormant states to persist through sub-optimal conditions in soils, which may be more frequent under disturbance. Although this had been discussed in earlier drafts, we overlooked its inclusion but were spurred to reintroduce it by this comment. We do not think that the timescales of bacterial generation are realistic under natural soil conditions. Bacterial generation times are often limited by local nutrient, especially carbon, availability and hard to study in situ; however, community turnover (replacement of 50% of the active population, which only constitutes ca. 90% of the total population) is estimated to take between 15 to 49 days and more than 130 days for fungi (Gunina et al., 2017).

Forth, the current manuscript touches very little on the different spatial scales that affect the different target organisms. Soil cores may integrate multiple habitats for soil mesofauna, but different “continents” for microbes. How will these different spatial (and also temporal scales; see point #3) affect the conclusions, such as in lines 348-352?

33. We think that our ordination results demonstrate that our sampling methods sufficiently address this query. With few outliers, all AVCs can be observed to cluster into distinct groups. This gives us confidence that our sampling approach and findings are sufficiently robust and representative across the large spatial scale of our sampling area.

Last and very minor suggestions: I suggest to add the taxa names to the different panels. Moreover, it would be very helpful to have information on the level of replication per habitat type in Figures 2 and 3.

35. We agree that adding names to the paneled figures makes the figures clearer and have made this change. The number of replicates per each AVC by organismal group has been added to Supplementary Table 9.

References

Black, H. I. J., et al. Assessing soil biodiversity across Great Britain: national trends in the occurrence of heterotrophic bacteria and invertebrates in soil. *J. Environ. Manage.* **67**, 255-266 (2003).

Bista, I. et al. Annual time-series analysis of aqueous eDNA reveals ecologically relevant dynamics of lake ecosystem biodiversity. *Nat. Comms.* **8**, 14087 (2017).

Bunce, R.G.H. et al. Vegetation of the British countryside – the countryside vegetation system. (Department of the Environment, Transport and the Regions, London, 1990).

- Chimienti, G., et al. Profile of microbial communities on carbonate stones of the medieval church of San Leonardo di Siponto (Italy) by Illumina-based deep sequencing. *Appl. Microbiol. Biotech.* **100**, 8537-8548 (2016).
- Dupont, A. Ö. C., Griffiths, R. I., Bell, T., Bass, D. Differences in soil micro-eukaryotic communities over soil pH gradients are strongly driven by parasites and saprotrophs. *Environ. Microbiol.* **18**, 2010-2024 (2016).
- Emmett, B. A. & the GMEP team. Countryside Survey Soils Manual. NERC/Centre for Ecology & Hydrology (CS Technical Report No. 3/07, CEH Project Number: C03259) (2008).
- George, P. B. L. et al. Evaluation of mesofauna communities as soil quality indicators in a national-level monitoring programme. *Soil Biol. Biochem.* **115**, 537-546 (2017).
- Griffiths, R. I. et al. The bacterial biogeography of British soils. *Environ. Microbiol.* **13**, 1642-1654 (2011).
- Gunina, A., Dippold, M., Glaser, B., Kuzyakov, Y. Turnover of microbial groups and cell components in soil: ¹³C analysis of cellular biomarkers. *Biogeosciences* **14**, 271-283 (2017).
- Gossner, M. M. et al. Land-use intensification causes multitrophic homogenization of grassland communities. *Nature* **540**, 266-269 (2016).
- Mahé, F. et al. Parasites dominate hyperdiverse soil protist communities in Neotropical rainforests. *Nat. Ecol. Evol.* **1**, 0091(2017).
- Maskell, L. C. et al. Exploring the ecological constraints to multiple ecosystem service delivery and biodiversity. *J. Appl. Ecol.* **50**, 561-571 (2013).
- Read, D. S. et al. Catchment-scale biogeography of riverine bacterioplankton. *ISME J.* **9**, 516-526 (2015).
- Smart, S. M., Robertson, J. C., Shield, E. J., van de Poll, H. Locating eutrophication effects across British vegetation between 1990 and 1998. *Glob. Change Biol.* **9**, 1763-1774 (2003).
- Tedersoo, L. et al. Global diversity and geography of soil fungi. *Science* **346**, 1256688 (2014).

Reviewers' Comments:

Reviewer #5:

Remarks to the Author:

The revised manuscript by George and colleagues is improved in many places, and I agree with several of the responses to reviewers' comments. Moreover, the text reads very well (despite some minor issues re punctuation)! However, I feel like some key criticisms have not been adequately addressed:

1. NMDS/CAP/ordinations: I do agree with the authors that NMDS broadly show similar trends in terms of separation of vegetation classes; however, looking more into the details of vectors indicates that there are some differences in the way that different environmental variables are correlated. Thus, I think that these differences are not negligible and likely important for the interpretation of the results. See e.g. the very nice discussion in lines 327-331, which supports my point. I do not agree with the authors that the CAP is more cluttered than NMDS – there are simple ways to improve readability, e.g. by slightly moving the names of environmental variables, using different colors etc. I also think that a table cannot fully substitute a figure, because the table does not show the reader how different environmental variables are correlated.
2. All reviewers have questioned the novelty of the paper, and now reading the authors' responses, I remain a bit concerned about this. The authors repeatedly refer to Gossner et al (2016) and argue that they are now testing relationships across more ecosystem types. Although I think this study indeed is very interesting from a soil ecological point of view, I am uncertain if these results justify publication in a high-profile journal. See e.g. text in lines 240-245. Given that this is the second revision, I sense that there is sufficient editorial support/interest.
3. I do not find response #33 very convincing. The authors only refer to an ordination result, while I was asking about potential spatial differences among taxa with very different sizes and spatial ranges. Since the authors claim that one novelty of the present study is the standardized assessment of these different groups, I do think that the differences in the size of these organisms should be discussed. In fact, in line 61, the authors directly refer to methodological challenges related to scale, but this is not followed up in the discussion. For example, are there any implications of this sampling in relation to species-area relationships?

Minor points:

4. What is different between the result given in line 199 in comparison to results presented in the section line 176-184?
5. L. 267/268: I do not think this sentence is clear enough.
6. L. 360-362: this is unclear to me.

Reviewer #5 (Remarks to the Author):

NMDS/CAP/ordinations: I do agree with the authors that NMDS broadly show similar trends in terms of separation of vegetation classes; however, looking more into the details of vectors indicates that there are some differences in the way that different environmental variables are correlated. Thus, I think that these differences are not negligible and likely important for the interpretation of the results. See e.g. the very nice discussion in lines 327-331, which supports my point. I do not agree with the authors that the CAP is more cluttered than NMDS – there are simple ways to improve readability, e.g. by slightly moving the names of environmental variables, using different colors etc. I also think that a table cannot fully substitute a figure, because the table does not show the reader how different environmental variables are correlated.

1. In response to this concern, we have run a complimentary suite of CAP ordinations and linear fitting of environmental variables for all organismal groups (lines 497-501). These figures and tables can be found in the Supplementary Material (Supplementary Tables 7-11; Supplementary Figs 6-10). We found that the R^2 values generated by linear fitting of environmental variables in both CAP and NMDS ordinations were highly correlated (Supplementary Fig 11) (lines 210-212 and 225-227). Therefore, we have included these findings in our Results but have not found sufficient grounds to alter our Discussion.

All reviewers have questioned the novelty of the paper, and now reading the authors' responses, I remain a bit concerned about this. The authors repeatedly refer to Gossner et al (2016) and argue that they are now testing relationships across more ecosystem types. Although I think this study indeed is very interesting from a soil ecological point of view, I am uncertain if these results justify publication in a high-profile journal. See e.g. text in lines 240-245. Given that this is the second revision, I sense that there is sufficient editorial support/interest.

2. Some of our findings confirm what has been found in other habitats (e.g. Gossner et al. 2016), but as highlighted previously, no other study has investigated in a uniform manner, this breadth of life, across so many habitats that are unconfounded by geography. Indeed, we share the Editorial interest and firmly believe that our insights into the ecology of the belowground biosphere is ideally placed in Nature Communications and importantly, will be highly cited.

I do not find response #33 very convincing. The authors only refer to an ordination result, while I was asking about potential spatial differences among taxa with very different sizes and spatial ranges. Since the authors claim that one novelty of the present study is the standardized assessment of these different groups, I do think that the differences in the size of these organisms should be discussed. In fact, in line 61, the authors directly refer to methodological challenges related to scale, but this is not followed up in the discussion. For example, are there any implications of this sampling in relation to species-area relationships?

3. *We now realise that our earlier response only conveyed our thoughts on the issue of spatial-scale. We have not analysed species-area relationships in this manuscript because it was not the aim of the present work to establish distribution patterns. Furthermore, we lack data on parts of the country (i.e. upland and alpine sites in mid-Wales) that we feel are needed to undertake that kind of analysis and so feel that such an analysis could be flawed. However, Reviewer 5 has raised an important issue with regards to the extreme range of body size in soil organisms. We acknowledge that our environmental DNA approach may not accurately cover an active animal community, however, it does capture a snapshot of the soil habitat by utilising environmental DNA (Creer et al., 2016; Deiner et al., 2017) left behind as animals travel through the soil medium. It is our opinion that this is a meaningful finding; though we have added lines 362-364 to address the problem of scale when collecting DNA from soil animals.*

What is different between the result given in line 199 in comparison to results presented in the section line 176-184?

4. *The results presented in lines (179-186) are a comparison of richness between organismal groups using Pearson's correlation coefficient, whereas the later result (line 199) details the direction of significant relationships between richness of each group and environmental variables. We have edited this line to make it clearer to the reader.*

L. 267/268: I do not think this sentence is clear enough.

5. *We have revised and expanded this sentence to better convey our message ("Richness of all microbial groups, except archaea, followed the land use productivity/management intensity gradient³² with higher richness in the highly productive and more disturbed grasslands and arable sites and lower richness in the least productive, relatively undisturbed upland Heath/bog sites." lines 272-275).*

L. 360-362: this is unclear to me.

6. *We have added to this line to make it clear to the reader that richness of most microbial groups declines along a gradient of decreasing land use productivity/management intensity (lines 370-372).*

References

Creer, S., Deiner, et al. The ecologist's field guide to sequence-based identification of biodiversity. *Methods Ecol. Evol.* **7**, 1008-1018 (2016).

Deiner, K., et al. Environmental DNA metabarcoding: transforming how we survey animal and plant communities. *Mol. Ecol.* **26**, 5872-5895 (2017).

Gossner, M. M. et al. Land-use intensification causes multitrophic homogenization of grassland communities. *Nature* **540**, 266-269 (2016).